# Motor, Physical, and Psychosocial Parameters of Children with and without Developmental Coordination Disorder: A Comparative and Associative Study

**DOI:** 10.3390/ijerph20042801

**Published:** 2023-02-04

**Authors:** Glauber C. Nobre, Maria Helena da S. Ramalho, Michele de Souza Ribas, Nadia C. Valentini

**Affiliations:** 1Department of Physical Education and Sports, Federal Institute of Education, Science and Technology of Ceará, Fortaleza 60020-181, Ceará, Brazil; 2Department of Physical Education, Federal University of Juiz de Fora, Juiz de Fora 36036-900, Minas Gerais, Brazil; 3Sports Center, Federal University of Santa Catarina, Florianópolis 88040-900, Santa Catarina, Brazil; 4School of Physical Education, Physiotherapy and Dance, Federal University of Rio Grande do Sul, Porto Alegre 90690-200, Rio Grande do Sul, Brazil

**Keywords:** motor coordination, self-efficacy, body mass index, strength, physical activity

## Abstract

(1) Background: Developmental coordination disorder (DCD) is a chronic impairment that affects several domains that mark the developmental trajectory from childhood to adulthood. Aim: This study examined the differences in physical and psychosocial factors for children with DCD and typical development (TD) and the associations between these factors with gross motor coordination. (2) Methods: Children with DCD (*n* = 166; age: M = 8.74, SD = 2.0) and TD (*n* = 243; Age: M = 8.94; SD = 2.0) attending private and public schools were screened using the MABC-2. Children were then assessed using the Körperkoordination test für Kinder (gross coordination), the Perceived Efficacy and Goal Setting System (self-efficacy), horizontal jump (lower limb strength), and dynamometer (handgrip strength). A semi-structured interview was carried out to examine the oriented physical activity practice in the daily routine, the time spent in the activities, and the use of public spaces to practice non-oriented physical activities. (3) Results: Children with TD showed scores significantly higher than children with DCD in almost all factors with small to very large effect sizes; the exceptions were self-care and daily physical activity. The structural equation model showed that for children with DCD, the BMI explained negatively and significantly the motor coordination (b = −0.19, *p* = 0.019), whereas physical activity (b = 0.25, *p* < 0.001), lower limb strength (b = 0.38, *p* < 0.001), and perceived self-efficacy (b = 0.19, *p* = 0.004) explained it positively. For children with TD, the BMI explained negatively and significantly the motor coordination (b = −0.23, *p* = 0.002), whereas physical activity (b = 0.25, *p* < 0.001) and lower limb strength (b = 0.32, *p* < 0.001) explained it positively. (4) Conclusions: The authors extended previous research by providing evidence that factors affecting motor coordination vary across childhood for children with DCD and TD. Self-efficacy was relevant only in explaining motor coordination for children with DCD.

## 1. Introduction

Developmental coordination disorder (DCD) is a neurodevelopmental condition characterized by severe difficulties, slowness, or inaccuracy in performing daily living motor skills [1,2]. These motor impairments may occur heterogeneously, so children with DCD may face different deficits in all activities, ranging from essential self-care and balance movements to more complex activities such as fine and-/or gross motor plays and schoolwork [2,3,4]. Notwithstanding that the etiology of this disorder is not yet precisely known, it has been believed that DCD may have a genetic component and is associated with low birth weight, preterm birth, and perinatal complications at birth; additionally, it is more common in males than in females [2,5]. The prevalence of DCD varies according to how the selection criteria are applied, but the literature frequently quotes an estimate of 5 to 6% of the pediatric population [2], which reflects a highly prevalent neurodevelopmental disorder that deserves closer attention.

Evidence has shown that compared to their typically developing peers, children with DCD are potentially at greater risk of physical inactivity [1,4] due to the successive difficulties and embarrassments experienced when performing motor tasks [6]. This lack of motor experience, added to the lower desire for engagement and social interactions, may become a risk factor for adverse health outcomes in childhood and throughout life [1,2,6,7]. In this sense, DCD has been associated with overweight and obesity [8], low levels of physical activity or participation in free and organized play and sport [1,5,9], poor fitness levels [1], and a reduced sense of self-concept [6,7,9]. These parameters are associated with a series of negative consequences [1,5,8], so it is possible to predict that when these factors coexist, the impact can be exponential, which leaves children with DCD at a considerable disadvantage in their developmental and health trajectories [1]. Previous studies have evidenced this detrimental interaction as a negative feedback loop, since poor motor coordination influences, and is also influenced by, physical, motor, and psychosocial markers [1,4]. Additionally, this undesirable cycle exposes children with DCD to a series of stressful situations, repeated frustration, failure, and bullying, which can have a substantial emotional impact on their perceived competence and conceptions of self, such as self-efficacy [6,7,10].

Self-efficacy is a psychological construct defined as an individual’s judgment about his/her capabilities to successfully perform different tasks of daily living [7,9,10,11]. Studies have shown that children with DCD frequently credit themselves as less competent, evidencing a lower sense of self-efficacy in different domains (self-care, schoolwork, leisure) and lower engagement and enjoyment in physical activities and physical education classes when compared with their typical development peers [6,7,10]. Even more, these children potentially show less social involvement and low perceived competence in cognitive, social and-/or motor dimensions, which can result in isolation, stigmatization, and problems regarding mental well-being [6,11]. These findings may be a warning sign since personal resources such as good feelings about self-efficacy, success, and confidence may offer motor and psychosocial benefits and attenuate the stress children with DCD regularly face [2,6].

As previously evidenced, DCD is a chronic impairment that has an echo in several domains that mark the developmental trajectory from childhood to adulthood. In this sense, several studies have examined the independent associations of physical [8], motor [1,12], and psychosocial [6,7] parameters with DCD. However, little is known about the combined association of these factors with DCD. The existing research studies have examined the concurrent comorbidities and the physical, motor, and social deficits related to the diagnosis of DCD but, in general, isolated rather than focusing on the effect of those factors combined to help explain motor outcomes. In particular, very little research has focused on muscle strength, which is critical to performing motor tasks. Further, incorporating a child’s views about their self-efficacy, specifically in performing different tasks regarding self-care, schoolwork, and leisure activities, can enhance the knowledge about the motor difficulties faced daily by children with DCD and provide information about functioning and perceptions of their competence in domains that could aid the implementation of interventions. This comprehensive range of factors will provide better insight and more reliable information about DCD-related factors, which may support motor and self-concept interventions for this broad and multifactorial disorder.

Therefore, this study extended previous research by investigating the combined effects of children’s muscle strength and self-perceptions on the motor coordination of children with and without DCD. This study had two aims: First, examine the differences between physical, behavioral, and psychosocial parameters and gross motor coordination of children with DCD and typical development (TD). Second, examine the relationships among BMI, physical activity, lower and upper limb strength, and perceptions of self-efficacy with motor coordination (outcome) of children with DCD and typical development aged 6 to 12 years. It was hypothesized that (1) children with DCD would show higher BMI and lower scores on motor coordination, upper and lower limb strength, physical activity, and perceptions of self-efficacy compared with children with typical development, (2) BMI would be inversely related to motor coordination, and that (3) perceptions of self-efficacy, strength, and physical activity would be directly related to motor coordination.

## 2. Materials and Methods

### 2.1. Participants

The sample was composed of 409 children (195 girls, 47.7%) 6 to 12 years old (M age = 8.86; SD = 2.01), attending private (20%) and public (80%) schools in Santa Catarina, a state in southern Brazil. A posteriori power analysis for sample size was conducted. For the sample of 409 children, the post hoc power analysis for structural equation model (SEM) was conducted for each group of children. Thus, considering the actual sample size (TD *n* = 243; DCD *n* = 166), the RMSEA effect size and degrees of freedom observed in SEM analyses, and an alpha (α) error probability of = 0.05, the power values observed were 0.82 and 0.81 for TD and DCD models, respectively; therefore, the sample size was appropriated for the present study. The inclusion criteria were children that were (1) 6–12 years of age, (2) currently attending public or private schools, and (3) had typical cognitive performance. Exclusion criteria were the presence of any physical and-/or cognitive impairment or a severe neuropsychiatric, neurological, and-/or chronic disorder (i.e., autism spectrum disorder, attention-deficit/hyperactivity disorder, learning disorder, generalized anxiety disorder, epilepsy, and musculoskeletal disorder).

Children with motor impairments and with typical development were included in the current study. All children were screened using the Movement Assessment Battery for Children-Second Edition (MABC-2) [13]. A cutoff below the 16th percentile confirmed the presence of a motor deficit on the MABC-2 (Criterion A-APA [14]); typical development (TD) was motor performance above the 16th percentile. Indirect measures were used to apply criteria B, C, and D for DCD. The children’s teachers provided information if the motor delays observed in each child meaningfully interfered with their daily activities (Criterion B-APA [14]). The information regarding children’s delays in achieving the motor milestone was used to verify if the onset of symptoms for the children in the DCD group and was obtained directly from the children’s school records and teachers’ reports of parents’ early school interviews (Criterion C-APA [14]). None of the children had cognitive impairments or learning disabilities, as reported by parents and teachers on the child school profiles (Criterion D-APA [14]).

Therefore, of the total sample, 166 children (66 girls, 39.8%; age: M = 8.74, SD = 2.0) performed below the 16 percentile and met all of the other criteria recommended by APA, 2013 [14], thus composing the group with DCD, while 243 children (129 girls, 53.1%; age: M = 8.94; SD = 2.0) composed the group with TD. The children were from low- to middle-class families with incomes ranging from one to ten times the Brazilian minimum wage (approximately USD 230 to 5300).

### 2.2. Instruments

*Group motor impairment screen.* The Movement Assessment Battery for Children—second edition (MABC-2), validated for Brazilian children [15], was used to screen children regarding the levels of motor impairment; as previously mentioned, the cutoff adopted for the motor deficit was a score below the 16 percentile, and for children with TD, a score above the 16th percentile. The MABC-2 has eight motor tasks that measure manual dexterity, ball skills, and balance with specific tasks for three age bands (AB1: 3–6 years; AB2: 7–10 years; AB3: 11–16 years); the three age bands were covered in the present study. Children were assessed individually at the schools by two trained assessors (master’s students with a previous degree in physical education); the test took 25 to 30 min for each child and was conducted in a quiet room.

*Motor coordination.* We used the Körperkoordinations-test für Kinder (KTK) [16] validated for Brazilian children [17] to assess children’s motor coordination. The KTK is composed of four motor tasks: (a) walking backward along a balance beam (WB); (b) jumping sideways over a slat (JS); (c) hopping for height on one foot (HH), and (d) moving sideways on boxes (MS); motor quotient was used.

*Perceived Efficacy and Goal Setting System—PEGS.* The children’s self-perceptions regarding their competence in everyday motor activities were assessed using the PEGS [18] adapted for Brazilian children [11]. The PEGS, a Likert scale, consists of 27 pairs of cards depicting a child in everyday activities, with a figure showing a child performing a particular activity with more competence and others with less competence. The figures depicted self-care (ex: cutting food, getting dressed), school/productivity (ex: finishing schoolwork, painting/writing), and leisure (ex: catching balls, riding a bicycle) activities. The child is presented with each pair of test cards and asked to select which card is more like him/herself. Afterward, the child indicates if the condition chosen is “very” or “a little” like him/herself, resulting in a four-point scale (1 = a lot like the better performance to 4 = a lot like the poor performance). PEGS was administered individually by schoolteachers and in a quiet room. The self-care, school/productivity, and leisure scores were obtained from the sum of individual items.

*Daily Physical Activity and Screen Routine*. An indirect measure of physical activity was used. A questionnaire about children’s daily active routines [19] was adapted to the present study. The questionnaire has been used previously in Brazil [20,21] with adequate internal consistency and reliability (Berleze and Valentini, 2022). The instrument has questions with multiple choices for answers, organized in five dimensions related to (1) the child’s means of transportation, (2) physical spaces for play, (3) frequent play activities, (4) the child’s leisure time with friends, and (5) administration of the child’s time in different activities. For the present study only, the questions were adapted to cover different forms of guided and unguided activities, and in the last dimension of the questionnaire, the time spent in each activity and the days in extracurricular activities were also assessed. Therefore, the questionnaire had questions related to (1) the child’s transportation from home to school (i.e., car, bus, bicycle), (2) public physical spaces for the child to actively play during leisure time (i.e., parks, backyards, wasteland urban areas, at school after school periods), (3) kinds of unguided activities to play and the use of a screen in leisure time (i.e., ball games, jump hoop, riding bicycles, tag games, dancing, television, computer/tablet/cellular phone), (4) practice in guided physical activities (i.e., extracurricular sports, gymnastics, dancing, and fighting lessons), and (5) the approximate time children spend daily on all unguided activities and the number of days and hours enrolled in guided extracurricular physical activity. The daily physical activity questionnaire was administered in quiet classrooms and sports gymnasiums with the support of the examiner.

*Upper limb strength.* A JAMAR manual hydraulic dynamometer (Hydraulic Hand Dynamometer^®^)-Model PC-5030J1, JAMAR, Miami, FL, USA, was used to assess children’s handgrip strength. The dynamometer was positioned with the elbow flexed at 90 degrees and the forearm in semi-pronation. It was adjusted according to the size of the child’s hands, positioned on the second phalanges of the index, middle, and ring fingers (American Society of Hand Therapists, 1992). Three attempts for each hand were assessed with a 60 s interval between them; the best result of the three attempts was used in the present study [22].

*Lower limb strength*. The horizontal jump was used to assess the strength of the lower limbs [23]. The children were instructed to jump as far as possible from standing with both feet in parallel. Three attempts for each child were assessed, and the best jump for distance was used. 

*Body Mass Index—BMI*. Body mass was assessed using a portable digital scale Welmy (200 kg/50 g), and a portable stadiometer (2 mt/.50 cm) was used to measure children’s height. BMI was computed using the standard formula [body mass (kg)/ height (m^2^)]. Anthropometric measurement followed the recommended protocol [24].

### 2.3. Procedures

The study was approved by the Human Research Ethics Committee of the State University of Santa Catarina (CAAE 45051215.8.0000.537) in southern Brazil. Participants were informed about the research procedures by the researchers in meetings held by the researcher team with the board of education, school administrator and teachers, and parents according to their schedules. The meetings were held in the school facilities. All children’s parents and the teachers enrolled in the study signed the informed consent. Each child provided verbal assent, agreeing to participate in the study. All participants were informed about the data confidentiality and privacy of personal information according to ethical guidelines. Further, for the statistical analysis, the dataset was codified, excluding the name of participants, and identification numbers were used. For all participants, it was ensured that they could revoke their consent and decline study participation at any time.

Four trained assessors with at least two years of experience were enrolled in data collection. Two assessors conducted the MABC-2 and the KTK; the other two administered the questionnaires and physical fitness tests. The test administration was conducted on non-consecutive days and organized into three ordered blocks. First, the MABC-2 test was administered for screening the children. Second, KTK, PEGS, daily physical activity, and the screen routine instruments were administered. Third, the BMI and upper and lower limb strength tests were conducted. All assessments were conducted at each child’s school. The duration of each assessment was approximately 20 min.

One assessor administered the MABC-2 and KTK tests in all children, and a second assessor independently scored 25% of each group (DCD *n* = 41; TD *n* = 60); intraclass correlation coefficient (ICC) confirmed the intra-rater reliability for the MABC-2 and KTK (ICC values of 0.90 and 0.94, respectively). A retest was conducted in 20% of the sample (DCD *n* = 33; TD *n* = 49) within a 10 day interval, for the MABC-2, KTK, and upper and lower limb strength tests; ICC results confirmed the inter-rater reliability (ICC values of 0.90, 0.92, 0.91 and 0.92, respectively)

### 2.4. Data Analysis

Means, standard deviations, and Pearson coefficient correlations were presented; independent sample *t*-tests were used to compare the groups of children with DCD and TD on BMI, physical activity, lower and upper limb strength, perceptions of self-efficacy, and gross motor coordination. Cohen’s D was used as the index of effect size (d: very small = 0.01, small = 0.20, medium = 0.50, large = 0.80, very large = 1.20). Structural equation modeling (SEM) was used to examine the plausibility of a linear causal relationship between physical, behavioral, psychosocial, and motor coordination factors. The initial model included one latent predictive variable with respective manifest variables: self-efficacy (self-care, leisure, and school/productivity); and another four manifest variables (BMI, physical activity, upper limb strength, and lower limb strength) and one latent outcome variable: motor coordination, with respective manifest variables: walking backward along a balance beam, jumping sideways over a slat, hopping for height on one foot, and moving sideways on boxes.

The maximum likelihood estimation method was utilized [25]. Multivariate outliers were examined using Mahalanobis squared distance (D²), and the normality of the data using the skewness and kurtosis univariate and multivariate with values >3 and >7.0 adopted as a violation of normality. Several indexes were used to assess the fit of the model—the comparative fit index (CFI) and the Tucker Lewis index (TLI); a value of 0.90 was considered as a minimum to infer model adjustment for CFI and TLI [26]. The root mean square error of approximation (RMSEA) with a 90% confidence interval (CI 90%) was also used; values lower than 0.05 were adopted as adequate [26].

A multigroup analysis was used to verify if the model was invariant for groups (DCD and TD). We conducted a configurational invariance analysis to determine if the number of dimensions and items in each dimension were acceptable for children’s groups [27]. The SEMs were conducted using MPLUS version 7.

## 3. Results

The skewness and kurtosis univariate and multivariate tests showed a normal distribution; values from 0.90 to 2.5 were found. Descriptive statistics, group comparisons, and correlations by group are presented in Table 1.

Children with TD showed higher levels of performance and self-efficacy than children with DCD in almost all factors with small to very large effect sizes; groups were similar regarding self-care and daily physical activity.

Regarding correlations, small to moderate, positive, and significant correlations were found between (1) motor coordination (walking backward, jumping sideways, moving sideways) and lower limb strength, (2) motor coordination (jumping sideways, moving sideways, and hopping for height) and upper limb strength, (3) motor coordination (walking backward and jump sideways) with physical activity, (4) motor coordination (moving sideways) and self-care (5) school and leisure self-efficacy and lower limb strength. Moderate to strong, significant, but negative correlations were found between (1) BMI and motor coordination (walking backward and moving sideways), (2) BMI and lower limb strength, and (3) BMI and school and leisure self-efficacy. The other correlations were non-significant or non-relevant to the study objectives (instrument item correlations).

Concerning SEMs, initially, the proposed model showed inadequate fit indexes (CFI = 0.78, TLI = 0.80, RMSEA = 0.09). Thus, the model was re-specified. Jumping sideways over a slat showed an inappropriate factorial load (l = 0.20) and was excluded from the model. In addition, upper limb strength did not significantly explain gross motor coordination (β = 0.08, *p* = 0.476) and, therefore, was also excluded. The new indexes showed a better adjustment (CFI = 0.97; TLI = 0.92, RMSEA = 0.02). After the re-specification, a multigroup analysis was conducted for each DCD and TD group.

For children with DCD, the model showed adequate adjustment indexes (CFI = 0.97, TLI = 0.91, RMSEA = 0.02). For this group, the BMI explained, negatively, and significantly, the motor coordination (β = −0.19 *p* = 0.019), whereas physical activity (β = 0.25 *p* < 0.001), lower limb strength (β = 0.38, *p* < 0.001), and perceived self-efficacy (β = 0.19, *p* = 0.004) together explained positively and significantly 27% the gross motor coordination variance (R = 0.27). Figure 1 presents these relationships.

For children with TD, the model showed poor adjustment (CFI = 0.90, TLI = 0.88, RMSEA = 0.06). The analysis showed a non-significant association between perceived self-efficacy and gross motor coordination (β = 0.08, *p* = 0.676). Therefore, self-efficacy was excluded from the model, and a better model adjustment was obtained (CFI = 0.94, TLI = 0.91, RMSEA = 0.03). For children with TD, the BMI explained, negatively and significantly, the motor coordination (b = −0.23 *p* = 0.002), whereas physical activity (β = 0.25 *p* < 0.001) and lower limb strength (β = 0.32, *p* < 0.001) together explained, positively and significantly, 21% of the gross motor coordination variance (R = 0.21). Figure 2 presents these relationships.

## 4. Discussion

This study first examined the differences between BMI, strength, physical activity, gross motor coordination, and self-efficacy of children with DCD and TD. Second, we examined how BMI, strength, physical activity, and self-efficacy were associated with motor coordination in children with DCD and TD.

### 4.1. Group Comparisons

Overall, the hypotheses were supported regarding differences between children with DCD and TD; children with DCD showed poor performance on gross motor coordination, upper and lower limb strength, and fragile perceptions of self-efficacy in school productivity and leisure activities compared to children with typical development. It is essential to highlight that the results showed that children with DCD performed significantly worse in all measures of gross motor coordination, which is essential to the acquisition of fundamental and specialized motor skills used in daily life physical activities and-/or sports [16,17]. Previous studies suggested that impairment in these tasks is a common outcome for children with DCD across infancy in different age groups [6,28,29,30]. These constraints may be related to sensorimotor and kinesthetic motor deficits, including poor postural control, which is crucial to balance, body orientation, and the development/performance of all motor skills [31,32]. Gross motor coordination impairments lead to restrictions in accomplishing daily motor tasks [3,14], restraining the child’s opportunities to play and interact; this may have a negative impact on his/her overall health outcomes [33].

Children with DCD also had lower scores on two measures of self-efficacy—school productivity and leisure activities. These results supported the contention that standard fragile self-perceptions are recurrent in children with motor disorders [6,7,9]. The results are aligned with two previous studies that examined self-efficacy using the same instrument—the Perceived Efficacy and Goal Setting System subscales. Engel-Yeger and Kasis [10] found that children with DCD, younger than the ones in the present study (5 to 9 years), had low perceptions of self-efficacy in leisure, school/productivity, and self-care, similarly to results reported by Nobre et al. [7] for children 6 to 8 years old with DCD and r-DCD. In the present study, no differences were found for self-care; children with DCD and TD had lower mean scores for activities such as cutting food, tying shoes, buttoning, getting dressed and zipping. However, children with DCD had more difficulties in tasks such as finishing schoolwork, painting, writing, organizing number playing sports and video games, catching balls, and riding a bicycle. It is crucial to note that school and leisure self-efficacy activities are conducted in social environments, exposing children with DCD to the observations and judgments of other children and adults—factors that, combined with their perceptions of low competence, could prevent them from interacting further and learning.

Children with DCD and TD were alike regarding PA, contrary to previous studies [1,4,5,9,28]. There is robust evidence to support that the higher the motor impairments among children, less engagement in physical activities is observed, leading to a predisposition for health problems among children with DCD [34,35]. This trend has been reported as the activity deficit hypothesis [4,36], and it was not observed in the present study. A non-direct measure was used in the present study; children answered a survey in an individual interview focused on the frequency and amount of time spent on physical activities rather than their intensity. This lack of a PA objective measure could explain the differences from previous studies.

Children with DCD had lower BMI than children with TD, contrary to the hypotheses and several previous studies [8,34,37]. For example, higher BMI, overweight, and obese prevalence have been reported for children with DCD [8,34] than for children with TD. Higher BMI has been related to lower engagement in PA in general for children [38]. Since no differences in PA were detected in the present study, a plausible explanation is that children with DCD may be moving enough to maintain adequate weight despite the coordination constraints.

### 4.2. BMI, Lower Limb Strength, Physical Activity, and Self-Efficacy Explain the Motor Coordination of Children with DCD and TD Differently

Regarding correlations, results demonstrated the close association and importance of gross motor coordination with the development of different biological, behavioral, and psychosocial factors during childhood [39]. Namely, children with higher scores in coordination tasks were the strongest (upper and lower limb strength), similarly to results reported previously [40,41]. Motor-coordinated children also had higher levels of PA; being motor-proficient reduced the chances of being physically inactive [39]. Additionally, those children demonstrated higher self-care. Children’s high self-perceptions and self-efficacy are related to higher motor scores [6,7,11,39]. The negative correlations between BMI and motor coordination and self-perceptions have also found support in the literature; children with higher BMI are less motor-competent and perceive themselves as less capable [42]. Lower limb strength was also negatively related to BMI; previously, it has been shown that children with obesity and overweight had strength deficits in lower-limb muscle groups [43] and were inhibited when they had to move their body mass against gravity [44].

Regarding the models, for children with DCD, the models showed adequate adjustment indexes, and the BMI significantly and negatively affected motor coordination. In contrast, physical activity, lower limb strength, and perceived self-efficacy explained positively and significantly the gross motor coordination; the study hypothesis was confirmed. The model had to be respecified for children with TD, and self-efficacy was excluded. Therefore, BMI explained motor coordination negatively and significantly, whereas physical activity and lower limb strength explained the variability in gross motor coordination positively and significantly. The hypothesis was partially confirmed.

Out of all the significant factors related to motor coordination, lower limb strength was the vital factor in the SEM model for children with DCD and TD. Previous cross-sectional studies showed strong evidence for the positive association between gross motor skills and muscle strength [45,46,47,48]. A similar trend has been reported in longitudinal studies. For example, Portuguese children who were stronger in standing long jump showed greater scores in gross motor coordination in time (age range 5 to 11 years) [37]. Moreover, two more longitudinal studies with Portuguese children evidenced that lower body muscular strength (also standing long jump) predicted motor coordination for 6- and 7-year-old children, respectively [49,50]. However, the relationship was only evident at some ages and for some motor skills; for example, standing long jump at 6 and 8 years of age did not predict ability in hopping or jumping sideways, moving sideways, or walking backward at 6, 7, and 8 years of age [50]; further, lower limb strength did not predict coordination in 8-year-old Belgian children [51]. Despite the levels of motor proficiency and the fact that children with DCD were outperformed by children with TD in strength tasks [28], this factor was critical in the model for both groups of children. This result is relevant, since it ratifies the importance of muscular strength in performing several daily activities and exercises, appearing as a powerful marker of motor and health aspects [52].

The second robust factor in both models was physical activity, similarly to the previous study with Brazilian children [38]. In addition, the results were also aligned with two previous systematic reviews that portraited data from around the world. The relationship between gross motor skills and physical activity has been demonstrated in the last decades with different magnitudes and age groups. For example, in a systematic review, Logan et al. [53], after describing the results from 13 studies, provided evidence of low to moderate relationships between gross motor skills and physical activity in early childhood and low to high relationships in middle to late childhood. Another systematic review, by Cattuzzo et al. [54], showed strong evidence for a positive association between gross motor skills and cardiorespiratory fitness. This evidence highlights the beneficial effect of higher physical activity on motor coordination. It also illustrates the need to develop programs that encourage children to adopt more active habits, since longitudinal information indicates that physical activity tends to decline during childhood [55].

The third factor in both models was BMI; for children with DCD and TD, the higher the BMI, the lower the motor scores. The results were aligned with a systematic review of 21 studies in children and adolescents (aged 3–18 years) that provided strong evidence for this inverse association with weight status [56,57] and were also consistent with longitudinal information [37]. In the same sense, D’Hont et al. [58] showed the disadvantageous effect of excessive weight on children’s motor coordination; however, the authors reported that this disadvantage increases in tests involving a greater proportion of body mass and/or when the body needs to move fast or against gravity; the most pronounced effect was observed with hopping for height). It is important to mention that in the present study, across ages, the relationship was small for both groups but remained in the models and explained the total variance of gross motor coordination. These results were contrary to the previous study with Brazilian children, also using similar data treatment, in which BMI failed to remain significant in the model [38]. Despite this contradiction, our results shed more light on the evidence of a negative relationship between excess weight and movement; children with overweight and obesity are more prone to spend a greater percentage of their time in sedentary activities and have motor delay [58].

In the present study, we provided further evidence that deserves special note. Out of all the significant factors related to motor coordination, self-efficacy was the only sign in the model for children with DCD; for TD children, this variable failed to remain in the model. The direct relationship was aligned with previous studies [59,60]. For illustration, recent systematic review findings indicated a relationship between poor motor skills and low self-esteem in eight studies. This relationship is more intense for children with DCD, as they repeatedly face motor difficulties and feelings of failure, which have significant emotional and well-being implications [6]. DCD is found to negatively impact self-concept. However, the magnitude of this relationship varies depending on age, gender, and co-morbidity, showing this construct’s complexity [60]. The lack of solid evidence portrays the difficulties of young children sometimes having difficulties being precise about their capabilities due to lack of experience and peer comparison of motor parameters [61]. It is essential to highlight that the present study extends previous studies by providing evidence for the critical role of self-efficacy in the motor coordination of children with DCD.

### 4.3. Strengths and Limitations

This study extends previous knowledge by investigating correlates of gross motor coordination in children with and without DCD. A further strength is that we examined these factors in an extensive age range with several factors, which enabled us to have a big picture of these factors across infancy. Indeed, our findings suggest that lower limb strength, a factor that is frequently disregarded, was the most robust in the models. We also provided evidence that correlates differ according to how motor-proficient the child is—self-efficacy was only relevant for the children with motor impairments. Thus, it is essential to consider the relevant evidence regarding the relationship between self-efficacy and gross motor coordination, specifically for children with DCD. The evidence of this study may guide researchers, teachers, and caregivers in implementing motor intervention programs that enhance DCD children’s self-confidence, feelings of belonging, optimal development, and participation in daily activities. A limitation of the present study was the lack of an objective measure of physical activity.

## 5. Conclusions

The reduced capability observed in motor coordination and strength and the lower scores on self-efficacy in children with DCD in the present study suggest that those children need supplementary support not only to improve motor skill performance but also to nurture their own self-perceptions. The results from the associative analysis suggested that not only motor impairments but also self-perceptions could impact their motor coordination. Participation in leisure activities, an extensive part of a child’s life, depends not only on being proficient but also on the child feeling confident to engage and play with playmates without fear of failure. Leisure activities demand social interactions and understanding of the different roles in the games to achieve a common goal—they may impose a high demand on children with DCD who lack confidence in their own abilities.

## Figures and Tables

**Figure 1 ijerph-20-02801-f001:**
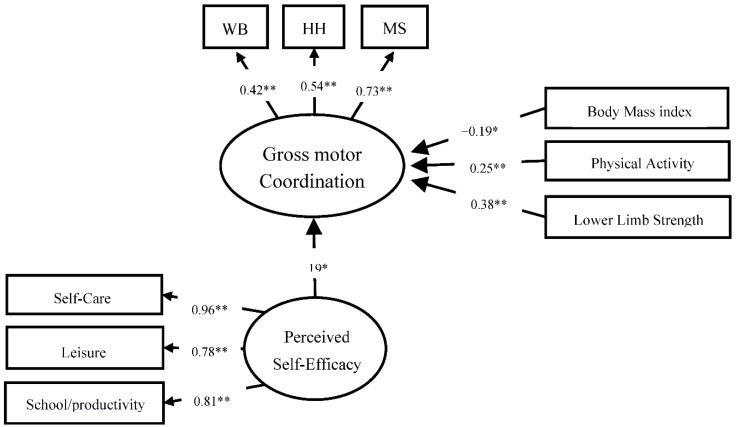
Children with DCD motor coordination model: BMI, PA, lower limb strength, and self-efficacy. Note: WB: walking backward along a balance beam; JS: jumping sideways over a slat; HH: hopping for height on one foot; MS: moving sideways on boards. * Statistically significant at *p* < 0.05. ** statistically significant at *p* < 0.001.

**Figure 2 ijerph-20-02801-f002:**
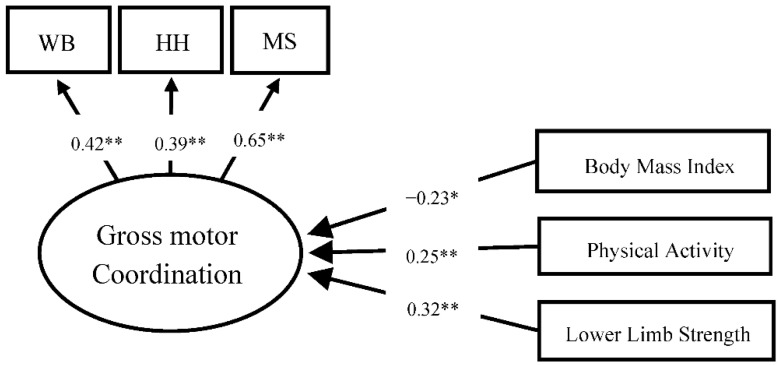
Children with TD motor coordination model: BMI, PA, and lower limb strength. Note: WB: walking backward along a balance beam; JS: jumping sideways over a slat; HH: hopping for height on one foot; MS: moving sideways on boards. * Statistically significant at *p* < 0.05. ** statistically significant at *p* < 0.001.

**Table 1 ijerph-20-02801-t001:** Descriptive statistics, comparison analysis between DCD group (*n* = 166) and TD group (*n* = 243), and correlations for the total sample (*n* = 409).

Variables	Group Comparisons M (SD)			Pearson Correlation Coefficient between Factors									
	Children with DCD *n* = 166	Children with TD *n* = 243	Cohen d	1	2	3	4	5	6	7	8	9	10
1-BMI (kg/m²)	16.9 (3.1)	23.8 (3.7) **	1.99 ^####^	-									
2-Lower limb strength (kg/f)	113.3 (16.7)	166.8 (25.0) **	2.44 ^####^	−0.74 **	-								
3-Upper limb strength (kg/f)	15.7 (5.7)	18.8 (7.0) **	0.48 ^#^	0.17 *	0.31 *	-							
4-KTK WB motor quotient	71.2 (9.4)	84.7 (8.0) **	1.57 ^####^	−0.52 *	0.56 **	0.05	-						
5-KTK JS motor quotient	73.7 (10.1)	86.6 (17.6) **	0.86 ^###^	0.04	0.15 *	0.29 *	0.28 **	-					
6-KTK HH motor quotient	107.1 (14.8)	112.6(15.7) **	0.45 ^#^	−0.02	0.01	0.34 *	0.29 **	0.21 **	-				
7-KTK MS motor quotient	64.6 (14.1)	76.6 (11.7 **	0.95 ^###^	−0.22 *	0.17 *	0.39 *	0.44 **	0.52 **	0.52 **	-			
8-PEGS self-care	10.1 (4.0)	9.6 (3.1)	0.14 ^#^	−0.04	0.11 *	0.08	0.00	0.07	0.13 *	0.18 *	-		
9-PEGS school productivity	17.6 (5.2)	12.8 (3.2) **	1.17 ^###^	−0.25 **	0.25 *	0.00	−0.10 *	−0.01	0.09	0.03	0.86 **	-	
10-PEGS leisure	24.1 (8.5)	18.3 (7.0) **	0.76 ^##^	−0.22 **	0.24 *	−0.02	−0.05	−0.01	0.10	0.08	0.85 **	0.92 **	-
11-Daily physical activity	39.9 (6.2)	42.0 (8)	0.29 ^#^	0.01	−0.05	0.14	0.15 *	0.42 **	0.09	0.34 **	0.10 *	0.05	0.05

Note. * *p* < 0.005; ** *p* < 0.001 significant differences at independent *t*-test and Pearson correlation (two tailed); Cohen’s d: ^#^ small = 0.20, ^##^ medium = 0.50, ^###^ large = 0.80, ^####^ very large = 1.20); WB: walking backward along a balance beam; JS: jumping sideways over a slat; HH: hopping for height on one foot; MS: moving sideways on boards.

## Data Availability

The data used in this study can be requested from one of the authors (N.C.V), who is the curator of the broad data set of this study.

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
