# Peer review of "Motor, Physical, and Psychosocial Parameters of Children with and without Developmental Coordination Disorder: A Comparative and Associative Study"

_ijerph, 2023, doi:10.3390/ijerph20042801_

Round 1
Reviewer 1 Report
1- The necessity of conducting research is not clear.
2- The methods section needs more details such as sample size calculation and ethical considerations.
3- The figures are not obvious and clear.
4- Application of the results needs more explanations.
Author Response
Comments and Suggestions for Authors
1- The necessity of conducting research is not clear.
Authors: a clearer reason to conduct the study is provided now – lines 86 to 100
2- The methods section needs more details such as sample size calculation and ethical considerations.
Authors: we added further information about the sample size calculation and ethical procedures
3- The figures are not obvious and clear.
Authors: we adjusted all figures
4- Application of the results needs more explanations.
Authors: We reinforced the application of the results.
Reviewer 2 Report
Include the aim of the study right before the Materials and methods.
Figure 1 and 2 needs checking spelling and grammar
Author Response
Include the aim of the study right before the Materials and methods.
Authors. The aim of the study is present now before Material e methods. We moved the previous last sentence to the middle of the paragraph.
Figure 1 and 2 needs checking spelling and grammar
Authors: we checked the spelling and grammar
Reviewer 3 Report
Dear authors,
I would like to thank you for the opportunity of reading your research. The research addresses the interesting topic of differences and associations between children with and without Developmental Coordination Disorder. In general it is a well written article that provides the needed information to the reader and that follows the rigour spected in a scientific publication. However, there are some minor issues that should be adressed before it can be accepted.
Abstract
Line 16: The text should not start with a contraction that has not been spelled completelly before.
Introduction
the introduction is clear. The state of the art, objectives and hypothesis are presented correctly.
Methods
Line 103: How do you decided the minimum sample needed to carry out the study? Is the statistical power of the study enough with the sample included? Please, it would be necessary to include the sample size calculation and estimated power for both groups.
Instruments
The instruments are well presented with high amount of details.
Procedures
Please, could you provide the order of the tests to be sure that there is no influence in the results? Was each evaluator in charge of some specific tests, or both of them performed all the test? Did you consider to calculate the inter-evaluator error?
Results
The results are well presented and provide enough information.
Discusion
Line 294: Should "tipical developement" be contracted?
Bibliography
Please, revise the bibliography in accordance of the journal instructions, as some errors have been detected (e.g. name of the journals)
Author Response
Comments and Suggestions for Authors
Dear authors,
I would like to thank you for the opportunity of reading your research. The research addresses the interesting topic of differences and associations between children with and without Developmental Coordination Disorder. In general it is a well written article that provides the needed information to the reader and that follows the rigour spected in a scientific publication. However, there are some minor issues that should be adressed before it can be accepted.
Authors: Thank you very much for the kind words and incentive. All your concerns were addressed
Abstract
Line 16: The text should not start with a contraction that has not been spelled completelly before.
Authors: we corrected this mistake
Introduction
the introduction is clear. The state of the art, objectives and hypothesis are presented correctly.
Authors: we thank you for the kind comment
Methods
Line 103: How do you decided the minimum sample needed to carry out the study? Is the statistical power of the study enough with the sample included? Please, it would be necessary to include the sample size calculation and estimated power for both groups.
Authors: we added further information about the statistical power of both groups
Instruments
The instruments are well presented with high amount of details.
Authors: we thank you for your comment
Procedures
Please, could you provide the order of the tests to be sure that there is no influence in the results? Was each evaluator in charge of some specific tests, or both of them performed all the test? Did you consider to calculate the inter-evaluator error?
Authors: we provided the order of application of the tests, and we inserted further information about the raters and intra-rater and inter-rater reliability (lines 212 to 238).
Results
The results are well presented and provide enough information.
Authors: we thank you for your comment
Discusion
Line 294: Should "tipical developement" be contracted?
Authors: We contracted typical development (lie4 353 now)
Bibliography
Please, revise the bibliography in accordance of the journal instructions, as some errors have been detected (e.g. name of the journals)
Authors: we revised the reference list